# Prussian Blue Nanoparticle-Entrapped GelMA Gels Laden with Mesenchymal Stem Cells as Prospective Biomaterials for Pelvic Floor Tissue Repair

**DOI:** 10.3390/ijms24032704

**Published:** 2023-01-31

**Authors:** Jirui Wen, Zhiwei Zhao, Fei Fang, Jingyue Xiao, Ling Wang, Juan Cheng, Jiang Wu, Yali Miao

**Affiliations:** 1Department of Obstetrics and Gynecology, Key Laboratory of Birth Defects and Related Diseases of Women and Children of MOE, West China Second University Hospital, Sichuan University, Chengdu 610041, China; 2Deep Underground Space Medical Center, West China Hospital, Sichuan University, Chengdu 610041, China; 3West China School of Basic Medical Sciences & Forensic Medicine, Sichuan University, Chengdu 610041, China

**Keywords:** pelvic organ prolapse, mesenchymal stem cells, gelatin–methacryloyl, Prussian blue nanoparticles, heat shock

## Abstract

Pelvic organ prolapse (POP) seriously affects elderly patients’ quality of life, and new repair materials are urgently needed. To solve this problem, we synthesized methacrylated gelatin (GelMA) hydrogels and incorporated photothermally active Prussian blue nanoparticles (PBNPs) to synthesize PBNP@GelMA. Then, MSCs were encapsulated in the PBNP@GelMA and exposed to a 1.0 W/cm^2^ of 808 nm laser for 10 min to perform heat shock pretreatment for the implantation of mesenchymal stem cells (MSCs). Next, we tested the repair efficacy of scaffold–cell complexes both in vitro and in vivo. Our results reveal that the heat shock treatment induced by PBNP@GelMA improved the viability of MSCs, and the underlying mechanism may be related to HSP70. Furthermore, 2 weeks after implantation in the SD rat model, the collagen content increased in the MSC implantation group and PBNP@GelMA implantation group. However, the muscle regeneration at the implanting position was mostly enhanced after the implantation of the heat-shock-pretreated MSCs, which illustrates that heat shock treatment can further promote the MSC-mediated muscle regeneration. Therefore, manipulating the cell environment and providing proper heat stimulus by using PBNP@GelMA with NIR is a novel strategy to enhance the regenerative potential of MSCs and to promote pelvic tissue repair.

## 1. Introduction

Pelvic organ prolapse (POP) is a common disease among aged women that seriously affects quality of life [1]. Transvaginal implantation of medical polypropylene mesh is a commonly used pelvic floor reconstruction method [2]. However, long-term adverse events associated with mesh (such as erosion, mesh exposure, pain, etc.) are difficult to deal with [3]. Therefore, new repair materials are urgently needed. At present, many researchers recognize that tissue engineering methods could significantly improve the therapeutic effect of POP surgery, and the application of mesenchymal stem cells (MSCs) in pelvic floor tissue repair has also received increasing attention. 

MSCs have been proven to stimulate tissue regeneration by regulating various biological processes related to tissue remodeling, such as angiogenesis, immune regulation, and cell recruitment and differentiation [4]. Altman et al. found that injected MSCs can enhance the vascularity of porcine acellular dermal matrix grafts in rats that have undergone inlay ventral hernia repair [5]. Furthermore, Li et al. observed that angiogenesis and collagen production increased after the implantation of MSCs into the abdominal wall of rats [6]. However, the lower survival rate of transplanted cells into host tissue significantly limits stem-cell-derived therapies [7]. Therefore, it is necessary to adopt material design strategies to promote cell viability. 

Hydrogels are gels of hydrophilic three-dimensional networks usually used as bioscaffolds for the delivery of drugs, extracellular vesicles, or MSCs. The gelatin–methacryloyl (GelMA) hydrogel is a kind of photosensitive hydrogel with various biomedical applications. Through its cell-attaching and matrix metalloproteinase-responsive peptide motifs, GelMA hydrogels can promote the proliferation and spread of cells [8]. As reported by O’Donnell BT, MSCs seeded on GelMA maintain robust cell viability after 28 days in growth medium [9]. On the other hand, manipulating the cell environment and providing proper stress stimulus is another strategy to enhance the regenerative potential of MSCs. Temperature is one such stressor that has recently been used to improve the vitality and function of MSCs. With exposure to heat shock treatment, the age-related alterations of MSCs were reduced, and the morphology of MSCs was maintained [10]. In another study, cells treated with heat shock achieved more doublings than cells in the control group, revealing that heat shock treatment also enhanced the proliferative potential of cells [11]. 

Prussian blue nanoparticles (PBNPs) are photothermally active nanoparticles that present an excellent biocompatibility. Under NIR laser irradiation (808 nm), PBNPs exhibit remarkable photothermal effects and heat release [12]. As our previous results, PBNPs did not significantly alter cell viability, proliferation, and migration activity in PBNP-labeled MSCs [13]. Herein, we synthesized GelMA hydrogels and incorporated PBNPs to synthesize PBNP@GelMA. Then, MSCs were encapsulated in the PBNP@GelMA. Under NIR laser irradiation (808 nm), PBNPs encapsulated in scaffold–cell complexes can release heat to perform heat shock pretreatment for MSCs. After the heat shock pretreatment, cell viability was detected in vitro. Additionally, using a rat abdominal wall model, in vivo experiments were carried out to evaluate tissue repair in the 14 d after the implantation of the preheated scaffold–cell complexes. Overall, these results suggest that scaffold–cell complexes hold great promise for pelvic floor tissue repair.

## 2. Results 

### 2.1. Structure and Properties of PBNP@GelMA 

To achieve heat shock pretreatment, PBNP@GelMA was prepared. As shown in Figure 1A, SEM observation indicated a porous surface morphology of PBNP@GelMA. Moreover, nanoparticles with rectangular structure could also be observed in the porous structure of PBNP@GelMA. The process of crosslinking of PBNP@GelMA is shown in Figure 1B, indicating the transition from a liquid state to a gel state. Tensile stretching tests and compression tests showed that PBNP@GelMA can withstand high tensile stress and compressive stress, indicating the satisfactory mechanical properties of PBNP@GelMA (Figure 1C,D). In addition, the compression modulus calculated according to the stress–strain curves was 38.32 ± 0.489 kPa. As shown in Figure 1E, when exposed to NIR, the PBNP@GelMA temperature increased to 40 °C within 5 min, indicating efficient photothermal conversion performance (Figure 1E).

### 2.2. Heat Shock Treatment Induced by PBNP@GELMA Promotes the Survival of MSCs

C3H10T1/2 is one of the best-characterized MSC lines. To evaluate the effect of heat shock treatment on the behavior of MSCs, NIR stimulations were performed for C3H10T1/2 encapsulated in PBNP@GelMA. As shown in Figure 2A, in the PBNP@GelMA+NIR group, the G1 phase fraction was decreased, and the G2/M phase fractions were increased, suggesting an elevated proliferation of MSCs. Furthermore, as shown in live/dead assays, NIR irradiation also affected cell viability (Figure 2B). After direct NIR stimulation, the viability of MSCs significantly increased. These results indicate that heat shock treatment can promote the cellular behavior of MSCs.

### 2.3. Role of HSP70 Expression in Elevated Cell Viability of MSCs Induced by Heat Shock Treatment

As shown in Figure 3A, a CCK8 assay revealed the elevated cell viability of MSCs induced by heat shock treatment. However, the main molecules involved in this event remain unclear. High expression of HSP70 proteins was detected (Figure 3B), suggesting that HSP70 was implicated in in elevated cell viability of MSCs induced by PBNP@GelMA. Therefore, we propose that heat shock treatment may affect MSCs through HSP70-initiated downstream transduction.

### 2.4. In Vivo Histochemical Assay Indicated Tissue Repair after the Implantation of Heat-Shock-Pretreated MSCs

As shown in Figure 4, HE staining showed that PBNP@GelMA was degraded after 2 weeks, and there were small neutrophils around the operative site, indicating the good histocompatibility of PBNP@GelMA. As shown in Figure 5, when compared to the control group, muscle bundles in the MSCs group and PBNP@GelMA group increased, suggesting that both the MSCs and PBNP@GelMA can promote muscle regeneration. However, the maximum muscle bundles occurred in the complex group with NIR, which illustrates that heat shock treatment can further promote MSCs-mediated muscle regeneration. As shown in Figure 6, the collagen content increased in the MSC group and the PBNP@GelMA group but not in the complex group with NIR. This result reveals that hyperplasia of fibrous connective tissue and scar repair were the main features in the MSC and PBNP@GelMA groups. Thus, more thorough tissue repair occurred in the complex group with NIR.

## 3. Discussion

Owing to a harsh environment at the host site, the low number or poor survival of cells after transplantation indicate that cell therapy does not reach the optimal level [14]. In this study, significant attention was paid to a strategy that simulates heat shock. We used a combination of hydrogels and photothermally active nanoparticles to improve stem cell function in order to maximize the therapeutic effect of stem cells. Our results indicate that under 808 nm NIR, the temperature of PBNP@GelMA increased, providing heat shock pretreatment for the MSCs. The MSCs treated with heat shock exhibited higher viability than the control group. Additionally, MSCs seeded in PBNP@GelMA significantly promoted tissue repair in the rat abdominal wall model.

As an ideal biomaterial, the 3D multipore structure of GelMA has an advantage in promoting cellular adhesion. As reported in a previous, the high water-absorbing ability of GelMA enhances the biocompatibility of the material and provides three-dimensional conditions [15]. In our study, the cell experiment revealed that PBNP@GelMA did not show poisoning on MSCs. Thus, PBNP@GelMA provided a suitable environment for the normal growth of MSCs.

The protective effect of mild heat stress on cell behaviors has been previously reported. Moloney TC found heat shock improved the survival of BMSCs by alleviating apoptosis [16]. Another study found that the expression of senescent-associated markers was significantly downregulated in MSCs after heat shock treatment, suggesting that heat shock can reduce the senescence of MSCs [11]. It should be noted that the differentiation of MSCs into different lineages was also improved after heat shock treatment. For example, in heat-shocked MSCs, osteogenic genes including ALP, osterix, ostepontin, etc., were upregulated [17]. MSCs exposed to heat shock also showed more oil red O uptake and expressed more adipogenesis markers after undergoing adipogenic induction [11]. Similarly, heat shock enhanced the chondrogenic differentiation of human MSCs by increasing expression of collagen type II and aggrecan [18]. Notably, the novel effects of heat shock include the promotion of wound healing [19]. It is clear that mild heat shock can affect the regeneration potential of MSCs in vitro, and these effects contribute to improved performance of these cells after transplantation. However, a correct method of application of heat shock in cell cultures is challenging. Incubators and water baths are not conducive to rapid warming of stem cells embedded in hydrogels. Thus, we used photothermally active nanoparticles to provide heat shock treatment, which achieved efficient photothermal conversion performance.

In this study, we found that the protective effects of heat shock can be attributed to elevated levels of heat shock protein HSP70. HSPs may function as molecular chaperones and can help to maintain normal cellular physiological activities and prevent negative protein interactions [20]. Studies indicate that HSPs can interact with various transcription factors, such as Nanog, Oct4, Sox2, and STAT3, and are therefore involved in various cell functions [21]. Therefore, alterations in the expression of HSPs directly affect stem cell characteristics, such as their proliferation capacity, as well as their vitality.

Our results also reveal that stem cell transplantation may promote tissue repair by promoting muscle regeneration, which is consistent with previous results. Several previous studies have shown that MSCs have great potential for multidirectional differentiation, including myogenic differentiation and angiogenic differentiation [22]. Konala et al. found that exosomes derived from MSCs contain various cytokines, which can nourish damaged skeletal muscle fibers and promote tissue repair in vivo [23]. The bioactive factors derived from MSCs can also remodel the ECM and inhibit tissue fibrosis [24]. This may explain why more thorough tissue repair occurred rather than scar repair in the group with heat-shock-pretreated MSCs.

## 4. Materials and Methods

### 4.1. Fabrication of GelMA

First, 20 g gelatin (Beyotime, Shanghai, China) was dissolved in 200 mL of PBS at 60 °C and thoroughly stirred until it was completely dissolved into a transparent yellow liquid. Then, 16 mL of MA (Beyotime, Shanghai, China) was added to the gelatin solution by a microinjection pump and allowed to fully react for 2 h. Next, 800 mL of preheated PBS was added to terminate the reaction, and the mixture was collected by suction filtration at 60 °C. Then, the obtained liquid underwent vacuum freeze drying for 48 h.

### 4.2. Fabrication of PBNP@GelMA

The photoinitiator LAP was dissolved in sterile PBS at a concentration of 0.25% (*w*/*v*) at 37 °C. Then, lyophilized foamy GelMA was dissolved in photoinitiator solution at 60 °C for 30 min to obtain a GelMA solution with a concentration of 10% (*w*/*v*). Next, the PBNP solution was added to the 10% (*w*/*v*) GelMA solution by a microinjection pump in order to prepare 1 μg/mL of a PBNP-entrapped GelMA solution. The mixed solution was filtered and stored at 4 °C. The PBNP@GelMA underwent ultraviolet irradiation for 5 min before being fully crosslinked.

### 4.3. Mechanical Testing of PBNP@GelMA

In order to characterize the mechanical properties of PBNP@GelMA, it was tested using an equivalence force test instrument. The samples were rectangular in shape with dimension of 1 cm width, 3 cm length, and 0.5 cm thickness, in order to test the stretching stress, and dimensions of 1 cm width, 2 cm length, and 1 cm thickness to test the compressive stress. The speed of stretching and compression was 1 mm/min. The compression modulus was calculated based on the stress–strain curves.

### 4.4. Photothermal Effect Evaluation of PBNP@GelMA

PBNP@GelMA was irradiated with 1.0 W/cm^2^ of 808 nm laser, and temperature changes were detected via a thermocouple thermometer.

### 4.5. MSCs Encapsulated in PBNP@GelMA

C3H10T1/2 is one of the best-characterized MSC lines; it was established from 14- to 17-day-old C3H mouse embryo cells. C3H10T1/2 MSCs were provided by Wuhan Pricella Biotechnology Co., Ltd., Wuhan, China. C3H10T1/2 was mixed with the polymer solution before forming hydrogels. Then, ultraviolet irradiation was used to encapsulate C3H10T1/2 in the PBNP@GelMA.

### 4.6. Cell Cycle Analysis

The MSCs were divided into three groups, with the cell density adjusted to 1 × 10^4^/mL: control group (cells inoculated and cultured in DMEM containing 10% FBS), PBNP@GelMA group (cells encapsulated in the PBNP@GelMA and cultured in DMEM containing 10% FBS), and PBNP@GelMA+NIR group (cells encapsulated in the PBNP@GelMA and cultured in DMEM containing 10% FBS with 1.0 W/cm^2^ of 808 nm laser for 10 min). Before cell cycle analysis, hydrogels were dissolved by GelMA lysate (EFL, Suzhou, China) first, and C3H10T1/2 were collected. Next, 10 mg/mL RNase and 1 mg/mL PI were used to dye the cells; then, the cells were detected on a flow cytometer.

### 4.7. Evaluation of Cell Viability

To determine the influence of heat activation on MSCs, live/dead assays were performed to assess the cell viability via staining with calcein and PI after 3 days of in vitro culture. Live cells were marked by green fluorescence, and dead cells were marked with red. Cells were imaged by fluorescence microscopy (Zeiss, Jena, Germany).

Furthermore, the cell viability was evaluated with a cell counting kit-8 (CCK-8). After starving for 24 h, 48 h, and 72 h, the 10 μL CCK8 reagent was diluted in 100 μL DMEM and added to each well. Then, the cells were incubated at 37 °C for an hour in the dark. The absorbance was measured at 450 nm.

### 4.8. Western Blot

The expression levels of HSP70 were quantitatively analyzed using Western blot (WB). Cells were disintegrated by 100 μL RIPA buffer (Beyotime, Shanghai, China) containing protease inhibitors and phosphatase inhibitors (Beyotime, Shanghai, China) at 4 °C. After 30 min, the samples were centrifuged at 10,000 rpm at 4 °C for 10min and quantified by an enhanced bicinchoninic acid (BCA) assay kit (Beyotime, Shanghai, China). Then, 30 μg of protein was electrophoresed in 10% SDS-PAGE gels and transferred to a polyvinylidene fluoride film. The protein samples were incubated with primary antibodies of GAPDH (1:5000, SAB) and HSP70 (1:1000, CST) at 4 °C overnight. The membranes were washed three times and incubated with secondary antibodies (1:1000; Sigma) for 2 h. Bands were visualized using a Gel Imager (ChemiDoc XRS+, Bio-Rad). The intensity of the blots was quantified with ImageJ software.

### 4.9. Surgical SD Rat Model

Forty female SD rats aged 8 weeks (225 ± 15 g) were provided by the laboratory animal unit of Sichuan University. Rats were raised under standard laboratory conditions and provided free access to food and water. Isoflurane was anesthetized by inhalation at 5 cc/min and maintained at 2 cc/min. After disinfection of abdominal skin three times, the external and internal oblique were removed to build a rat abdominal wall defect model with dimensions of 1 cm × 1 cm. Then, the MSCs, PBNP@GelMA solution, or 1.0 W/cm^2^ 808 nm NIR-pretreated complex (MSCs mixed in PBNP@GelMA solution) was injected to the injured site. Next, ultraviolet irradiation was used to change the in situ gel solution into a gel. After 2 weeks of healing time, the samples were taken for histological staining.

### 4.10. Histological Staining

The samples were fixed with paraformaldehyde for 24 h, dehydrated in an alcohol gradient, and placed on a paraffin slicer for sectioning. The slices were approximately 3–4 µm thick, numbered, and placed in a 60 °C oven for baking. HE staining, Masson trichrome staining and Sirius Red staining were performed according to the manufacturers’ instructions. Then, morphological observations were performed at random, and samples were photographed using a light microscope. The percentage of muscle area or collagen area was calculated in three randomly selected regions using ImageJ software.

### 4.11. Statistical Analysis

All statistical analyses were performed with SPSS 21.0 statistical software. All statistical data are shown as means ± standard error of the mean, and comparisons between two groups were performed using repeated measurement analysis of variance (ANOVA). The levels of statistical significance between two corresponding groups were set to *p* < 0.05 and 0.01.

## 5. Conclusions

We developed a heat shock strategy using GelMA hydrogels embedded with PBNPs for the implantation of MSCs. PBNPs in GelMA hydrogels can be excited by external NIR to achieve efficient photothermal conversion performance. In vitro experimental results with MSCs reveal that PBNP@GelMA-mediated heat shock treatment successfully promoted the viability of MSCs. In vivo research suggests that the heat activation of MSCs promotes tissue repair by promoting muscle regeneration. This approach is suitable for clinical application of MSCs for pelvic tissue repair.

## Figures and Tables

**Figure 1 ijms-24-02704-f001:**
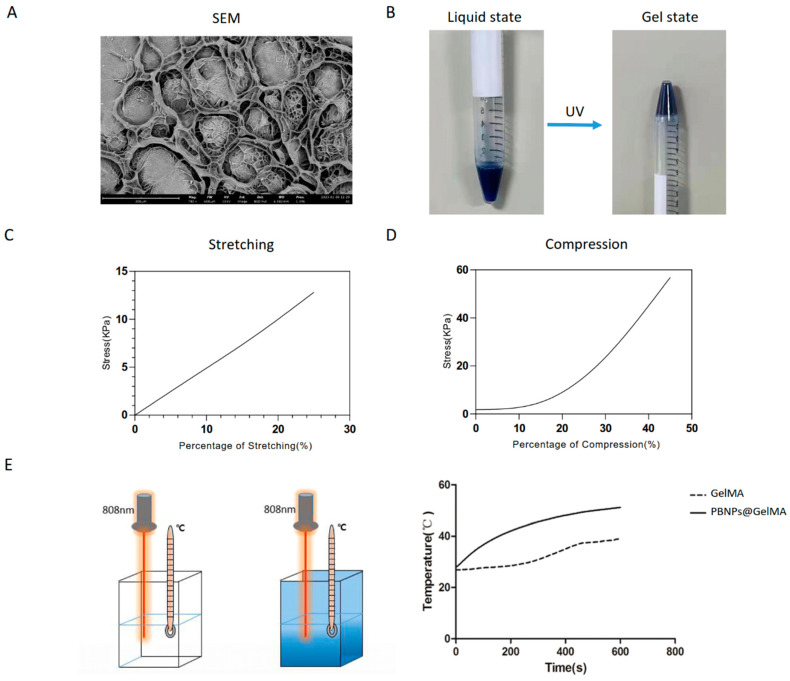
Characterization of PBNP@GelMA hydrogel. (**A**) SEM image. (**B**) The process of crosslinking. (**C**) Stretching test. (**D**) Compression test. (**E**) Temperature increase behavior. PBNP@GelMA was irradiated with 1.0 W/cm^2^ of 808 nm laser, and temperature changes were detected.

**Figure 2 ijms-24-02704-f002:**
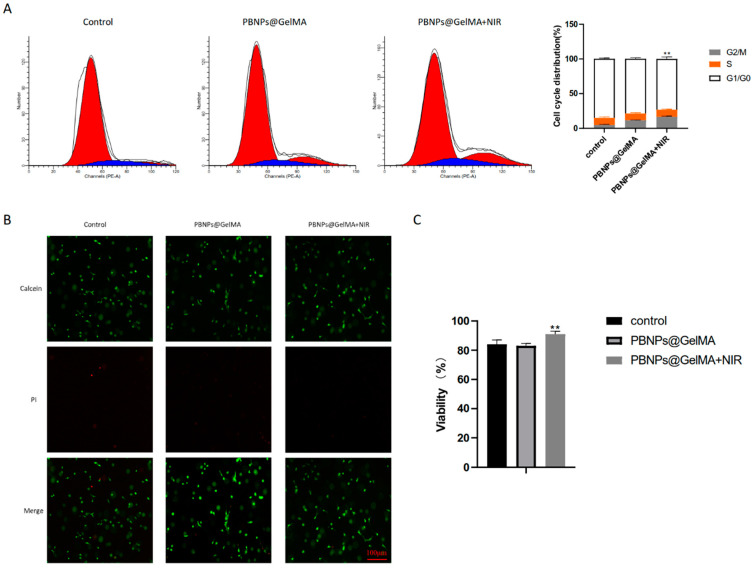
Heat shock treatment induced by PBNP@GelMA regulated the behavior of MSCs. (**A**) Flow cytometry assay to explore the cell cycle. (**B**) Fluorescence image of the Live/dead assays. Green fluorescent cells are alive, and red fluorescent cells are dead. (**C**) Viability calculated by Live/dead assays (*n* = 3). Samples without implants were used as a control group; ** *p* < 0.01.

**Figure 3 ijms-24-02704-f003:**
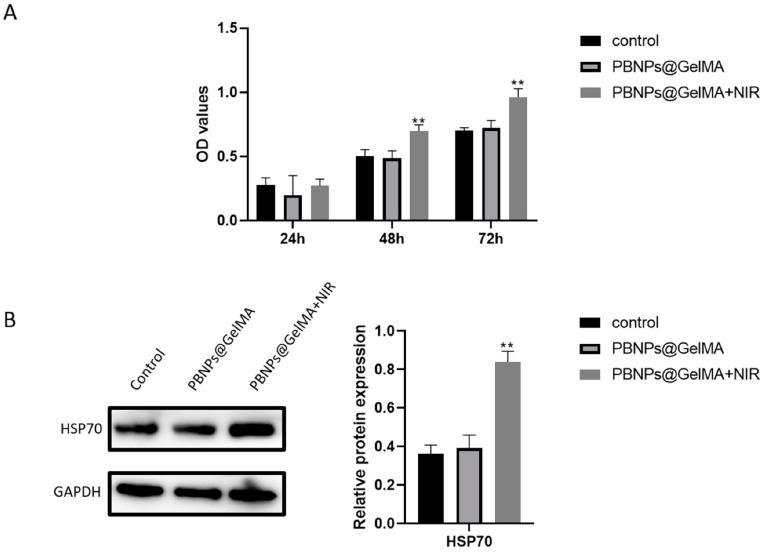
HSP70 played a key role in elevated cell viability of MSCs induced by heat shock treatment. (**A**) Viability of MSCs after 1, 2, and 3 days was shown by CCK8. (**B**) HSP70 expression in MSCs after heat shock treatment and associated statistics (*n* = 3). Samples without implants were used as a control group; ** *p* < 0.01.

**Figure 4 ijms-24-02704-f004:**
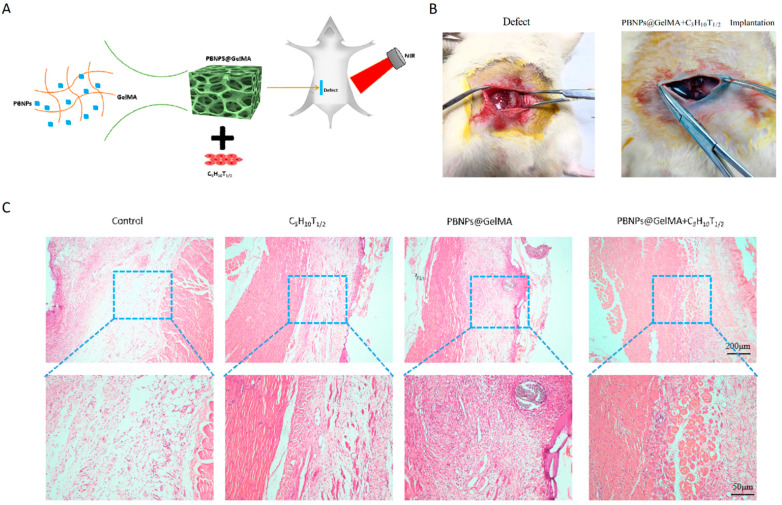
Good histocompatibility of PBNP@GelMA. (**A**) Schematic diagram of surgical repair process mediated by the implantation of heat-shock-pretreated MSCs. (**B**) General view of the injured site with or without the scaffold–cell complexes. (**C**) HE staining of abdominal wall muscle defect treated with control, MSCs, PBNP@GelMA, and MSCs + PBNP@GelMA 2 weeks after surgery.

**Figure 5 ijms-24-02704-f005:**
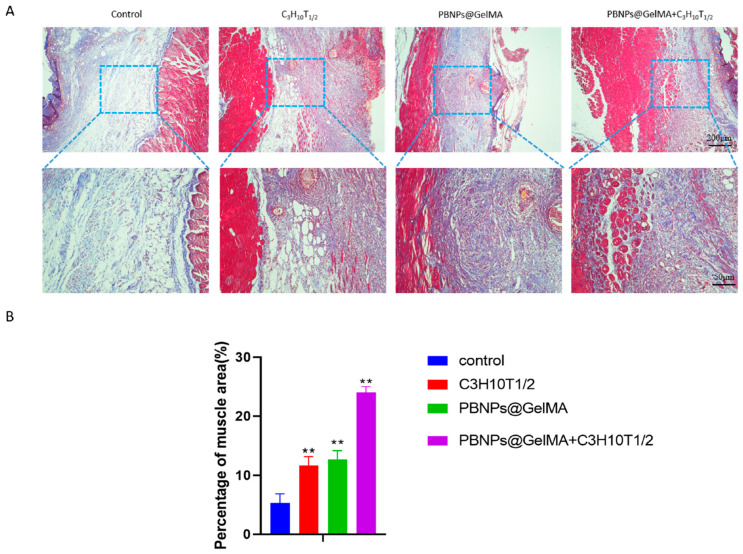
Observation of muscle regeneration with Masson staining in vivo. (**A**) Masson staining images of abdominal wall muscle defect treated with control, MSCs, PBNP@GelMA, and MSCs + PBNP@GelMA 2 weeks after surgery. (**B**) Statistics of percentage of muscle area (*n* = 3). Samples without implants were used as a control group; ** *p* < 0.01.

**Figure 6 ijms-24-02704-f006:**
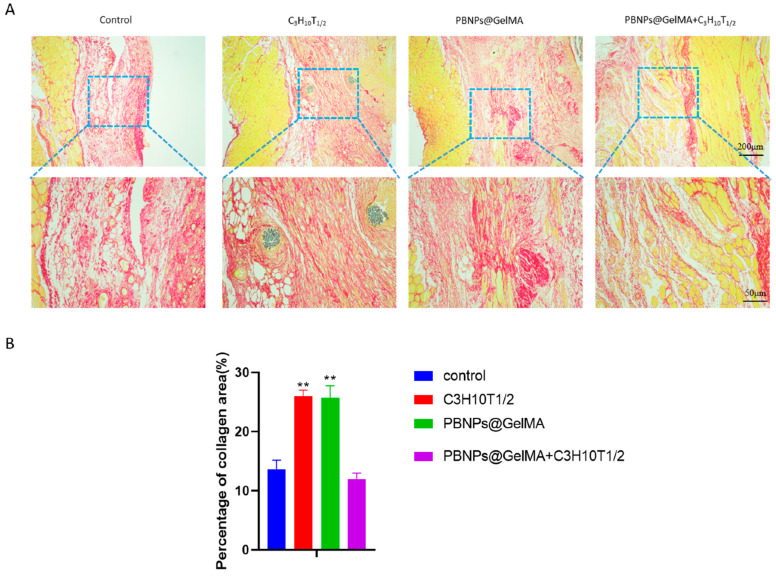
Observation of collagen regeneration with Sirius Red staining in vivo. (**A**) Sirius Red staining images of abdominal wall muscle defect treated with control, MSCs, PBNP@GelMA, and MSCs + PBNP@GelMA 2 weeks after surgery. (**B**) Statistics of percentage of collagen deposited after treatment (*n* = 3). Samples without implants were used as a control group; ** *p* < 0.01.

## Data Availability

The datasets presented in this study are available from the corresponding author upon reasonable request.

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
