# Peer review of "Prussian Blue Nanoparticle-Entrapped GelMA Gels Laden with Mesenchymal Stem Cells as Prospective Biomaterials for Pelvic Floor Tissue Repair"

_ijms, 2023, doi:10.3390/ijms24032704_

Round 1
Reviewer 1 Report
The scientific content of the paper is good, but I suggest you improve your English and pay attention to text formatting, punctuation, and spaces in the text. Here are some specific suggestions:
Line 15: "in vivo" and "in vitro" as all Latin words should be written in italics;
fig. 1 A: is it possible to magnify the figure? It isn't easy to see SEM parameters
fig.1 C e D should have the same dimension of character and the same resolution
fig. 2 A should have a better resolution.
Author Response
Dear Reviewer,
Thank you for your letter and comments on our manuscript titled “Prussian blue nanoparticles-entrapped GelMA gels laden with mesenchymal stem cells as prospective biomaterials for pelvic floor tissue repair”. And great thanks to the editors for giving us enough time to revise our paper. We have addressed the reviewers’ comments to the best of our abilities, and revised text to meet the requirements. We hope this meets your requirements for a publication. We marked the revised portions in red in the manuscript. The main comments and our responses are detailed below:
The scientific content of the paper is good, but I suggest you improve your English and pay attention to text formatting, punctuation, and spaces in the text. Here are some specific suggestions:
Line 15: "in vivo" and "in vitro" as all Latin words should be written in italics;
Author response: Thank you for your good evaluation and kind reminding, and we have made revisions according to the comments.
fig. 1 A: is it possible to magnify the figure? It isn't easy to see SEM parameters
Author response: Thank you for your kind reminding. We have used a new SEM image to replace the previous image to show the SEM parameters.
fig.1 C e D should have the same dimension of character and the same resolution
Author response: Thank you for your valuable suggestion. We have modified the fig.1 C, D to the same resolution.
fig. 2 A should have a better resolution.
Author response: Thank you for your valuable suggestion. We have modified the fig. 2 A to a better resolution.
Reviewer 2 Report
This research paper by Jirui Wen et al. reported the delivery of adipose-derived stem cells (ADSCs) using nanoparticles-entrapped GelMA gels for treatment of pelvic floor tissue repair. The authors synthesized GelMA hydrogels and loaded PBNPs into the hydrogels. Following that, the authors characterized the morphology, mechanical strength, and thermo-responsive behavior of the composite hydrogels. Then the authors evaluated the biocompatibility of their hydrogel systems showing that heat shock in the PBNPs@GelMA promoted cell survival in vitro and tissue regeneration in abdominal wall muscle defect model. Although this manuscript presented some interesting results, a lot of errors in current manuscript significantly lowered the quality of the manuscript.
1. Line 16, What is PBNPs? Please add the full name the first time using an abbreviation.
2. Line 45, “GelMA hydrogel is a kind of photosensitive hydrogel”. GelMA is a photo-crosslinkable polymer but what does photosensitive hydrogel mean here?
3. Figure 1, panel A) legend says TEM image but the results say SEM image. The image clearly shows a SEM image. Panel B) in the image is missing. Please add more text in the panel E) legend to explain the scheme.
4. Line 74, line 76, Fig.c,d; Fig.e?
5. Figure 1, what is the compression modulus for your hydrogel materials?
6. Figure 2, panel B is missing scale bars.
7. Figure 4, panel and legend does not match.
8. Line 107-108, what happened to the cells after two weeks?
9. What is the C3H20T1/2 group in your Figure 4? The authors showed this in their results but never mentioned a word about the annotation here.
10. Line 186, please add the detailed information about the reagents used. What chemistry was used to conjugate MA to gelatin? What method used to remove the byproducts?
11. Line 207, how did the author load the cells into the hydrogel? Were the cells mixed with the polymer solution before forming hydrogels or cells were seeded on top of the hydrogels?
12. Line 218, Please add more information about the protein extraction and how much total protein was used for gel electrophoresis.
Author Response
Dear Reviewer,
Thank you for your letter and comments on our manuscript titled “Prussian blue nanoparticles-entrapped GelMA gels laden with mesenchymal stem cells as prospective biomaterials for pelvic floor tissue repair”. And great thanks to the editors for giving us enough time to revise our paper. We have addressed the reviewers’ comments to the best of our abilities, and revised text to meet the requirements. We hope this meets your requirements for a publication. We marked the revised portions in red in the manuscript. The main comments and our responses are detailed below:
- Line 16, What is PBNPs? Please add the full name the first time using an abbreviation.
Author response: Thank you for your valuable suggestion. We have added the full name of PBNPs (Prussian blue nanoparticles) when it was used at the first time.
- Line 45, “GelMA hydrogel is a kind of photosensitive hydrogel”. GelMA is a photo-crosslinkable polymer but what does photosensitive hydrogel mean here?
Author response: That was exactly true what the reviewer considered, we agreed with the reviewer's opinion and have deleted this part of the description.
- Figure 1, panel A) legend says TEM image but the results say SEM image. The image clearly shows a SEM image. Panel B) in the image is missing. Please add more text in the panel E) legend to explain the scheme.
Author response: Thank you for your valuable suggestion. We have modified the figure legend of Figure 1 according to the suggestions.
- Line 74, line 76, Fig.c,d; Fig.e?
Author response: Thank you for your kind reminding, and we have corrected it to Fig. 1c, d; Fig. 1e.
- Figure 1, what is the compression modulus for your hydrogel materials?
Author response: Thank you for your meaningful suggestion. We have calculated the compression modulus from the stress-strain curves, which was 38.32±0.489 kPa.
- Figure 2, panel B is missing scale bars.
Author response: Thank you for your kind reminding, and we added the missing scale bars.
- Figure 4, panel and legend does not match.
Author response: We were so sorry for the mistake, and we have corrected it.
- Line 107-108, what happened to the cells after two weeks?
Author response: That was exactly important what the reviewer considered. Tissue regeneration is a comprehensive and dynamic process that requires transplanted cells, cytokines and growth factors to interact, including cell migration and proliferation (A. C. Gonzalez, T. F. Costa, Z. D. Andrade, et al, Wound healing-A literature review, Anais brasileiros de dermatologia., 2016, 91, 614–620.). It is difficult to track the migration and proliferation of stem cells in vivo without the label of stem cells. In our manuscript, we found the heat shock pretreated MSCs were with better tissue repair ability, which suggested that heat shock treatment may improve the function of MSCs in vivo.
- What is the C3H20T1/2 group in your Figure 4? The authors showed this in their results but never mentioned a word about the annotation here.
Author response: The suggestion of the reviewer was valuable and we have added the description of the C3H20T1/2 group in the Methods part.
- Line 186, please add the detailed information about the reagents used. What chemistry was used to conjugate MA to gelatin? What method used to remove the byproducts?
Author response: Thank you for your meaningful comments. Our statement in the original text was not clear and we had corrected. In the 4.1, more details on the preparation of the GelMA were provided as follows:
“Firstly, 20 g gelatin (Beyotime, China) was dissolved in 200 mL of PBS at 60 °C and thoroughly stirred until it was completely dissolved into a transparent yellow liquid. Then, 16 mL of MA (Beyotime, China) was added to the gelatin solution by microinjection pump to fully reacting for 2 h. Next, 800 ml of preheated PBS was added for the termination of the reaction, and the mixture was collected by suction filtration at 60 °C. Then the obtained liquid underwent vacuum freeze drying for 48 h.”
- Line 207, how did the author load the cells into the hydrogel? Were the cells mixed with the polymer solution before forming hydrogels or cells were seeded on top of the hydrogels?
Author response: The suggestion of the reviewer was valuable. In our protocol, the cells were mixed with the polymer solution before forming hydrogels. We had stated this in the Methods description of the manuscript.
- Line 218, Please add more information about the protein extraction and how much total protein was used for gel electrophoresis.
Author response: The suggestion of the reviewer was valuable and we had stated in the Methods description of the manuscript as follows:
“Cells were disintegrated by 100μl RIPA buffer (Beyotime, China) containing protease inhibitors and phosphatase inhibitors (Beyotime, China) at 4℃. After 30 minutes, the samples were centrifuged with 10,000rpm at 4°C for 10min, and quantified by enhanced bicinchoninic acid (BCA) assay kit (Beyotime, China). 30 μg of protein were electrophoresed in 10% SDS-PAGE gels.”
Reviewer 3 Report
The Manuscript „Prussian blue nanoparticles-entrapped GelMA gels laden with 2 adipose-derived stem cells as prospective biomaterials for pelvic floor tissue repair” presents a research on gelatin-mathacryolyl hydrogels with incorporated Prussian blue nanoparticles synthesized to perform heat shock pretreatment for the implantation of adipose-derived stem cells. Based on the obtained results these biomaterials exhibited beneficial properties and behavior for the specific biomedical application. The results are clearly and understandably presented, and although the GelMA hydrogels are already investigated for such and similar application, this manuscript displayed the innovation and novelty by incorporating PBNPs for the heat shock pretreatment. According to the in vitro and in vivo studies, the treatment promoted the viability of ADSC and the tissue repair by promoting muscle regeneration. Thus, the manuscript can be published with minor revisions which refer to grammar check and first mention of the abbreviations (in Abstract).
Author Response
Dear Reviewer,
Thank you for your letter and comments on our manuscript titled “Prussian blue nanoparticles-entrapped GelMA gels laden with mesenchymal stem cells as prospective biomaterials for pelvic floor tissue repair”. And great thanks to the reviewer for your kind comment. As the reviewer suggested, we have corrected the grammar mistakes and added the first mention of the abbreviations in Abstract.
Reviewer 4 Report
In ‘Prussian blue nanoparticles-entrapped GelMA gels laden with adipose-derived stem cells as prospective biomaterials for pelvic floor tissue repair’, Wen et al. prepared GelMA hydrogels with Prussian blue nanoparticles to provide a heat shock treatment of adipose-derived stem cells. They examined the viability of the cells and the mechanism of action in vitro. They also examined muscle regeneration after implantation in vivo and noted improvements. While the study goal is clear, the manuscript lacks sufficient details about a number of the experiments that were performed, and complete characterization of the materials is lacking.
Specific comments:
- The abstract should be checked so that all abbreviations are written in full. Unless limited by word count, the abstract could be expanded to better explain the heat shock treatment and to better summarize the results obtained in the study.
- The abstract and the end of the introduction should be checked for consistency with the results. For example, the length of the in vivo study is unclear.
- The degree of modification of the GelMA should be measured, for example, by NMR.
- The mechanical characterization should be properly performed, including for example calculations of the modulus from the stress-strain curves. Replicate samples should be measured and results reported with standard deviations.
- There is a typo in the Figure 1a caption. This is showing a SEM image.
- Is it possible to see the nanoparticles in the SEM images?
- It is not clear if the cells are cultured on the surface or inside the hydrogels. This should be clarified.
- The interpretation of the calcein staining results is somewhat questionable. Conclusions about viability cannot be made unless there is also a measurement of the number of dead cells. An increase in cell number could indicate that the cells are proliferating more, not that less cells are dying.
- The different curves in Figure 2a are not labelled. It is not clear what these graphs are showing. The cell cycle data should be analyzed and quantitative results reported.
- Is a material that degrades in less than 2 weeks really expected to have an impact in the treatment of pelvic organ prolapse?
- In Figure 5b, more details need to be provided about how the percentage of muscle area is calculated. For example, is this looking at the entire defect or just selected images? Is this based on manual counting or some sort of image analysis. The number of independent replicates used in these studies needs to be reported (usually in the figure caption). Likewise for the collagen results reported in Figure 6.
- Are the C3H10T1/2 the adipose-derived stem cells? The C3H10T1/2 cell line is a fibroblast cell line and not an adipose-derived stem cell.
- In the discussion, the statement that the PBNPs@GelMA has better resistance to deformation is not supported by the experiments, as no comparisons were performed during the mechanical characterization.
- More details are needed in the materials and methods section to allow others to reproduce the experiments. In particular:
o Section 4.1 needs more details on the preparation of the GelMA and then how the hydrogels are prepared. For example, is a photo-initiator used in the reaction?
o Section 4.4 needs more details on how the cells are cultured (also the source of the cells) and how they are encapsulated in the hydrogels. For the cell cycle analysis, more information needs to be provided about how the cells were retrieved from the hydrogels.
o In section 4.5, calcein is not an immunofluorescence staining methods.
o In section 4.6, details should be provided on how the proteins are obtained (e.g., cell lysis protocol).
o In section 4.7, more information is needed about how the defect repair was done. For example, were preformed hydrogels implanted or were they formed in situ? Also, no information is provided about the healing time.
Round 2
Reviewer 2 Report
The authors have addressed all my comments.
Reviewer 4 Report
The authors have sufficiently addressed the reviewer comments.